# Soil Hydrology Process and Rational Use of Soil Water in Desert Regions

Zhongsheng Guo [1,2]

1. Institute of Soil and Water Conservation, Northwestern A & F University, Yangling, Xianyang 712100, China; guozs@ms.iswc.ac.cn; Tel.: +86-29-8701-2411
2. Institute of Soil and Water Conservation, CAS & MWR, Yangling, Xianyang 712100, China

**Abstract:** There is a balanced plant–water relationship in the original vegetation in the desert area. With the increase in the population and social development of the desert area, people need the goods and services of the forest vegetation ecosystem. To meet the growing demand for plant community goods and services, more original vegetation has been changed into non-native vegetation, such as in the Loess Plateau in China. However, with the plant growth, sometime soil drying happens and becomes gradually serious with time in most desert regions. Serious drying of soil eventually results in soil quality degradation, vegetation decline, and crop failure, which influence the produce and supply of forest vegetation goods and services in the market in dry years or waste of soil water resources in wet years, which wastes precious natural resources. In order to use soil water rationally, soil water must be used in a sustainable way and the plant–water relationship has to be regulated for the Soil Water carrying capacity for vegetation in the key period of plant–water relationship regulation to carry out a sustainable use of natural resources, high-quality sustainable development of forest and grass, and high-quality production of fruit and crops in desert regions.

**Keywords:** infiltration; run off; deep leakage; soil water; plant growth; use limit of soil water resource by plants; carrying capacity of soil moisture for vegetation; key period of plant–water relationship regulation; rational use of water resources



## 1. Introduction

Climate change will decrease precipitation and increase evaporation, causing severe and worldwide drought. There is a multitude of plant responses to water stress. Fast responses (acclimation) are quite well understood, e.g., closing of stomata, rolling the leaves, and increasing leaf angles [1,2]. Slow responses include earlier reproduction (before drought) [1], rigid cell walls or smaller cells to tolerate low water potential [3], and differences in rooting depth [4]. In water-limited regions, the soil water status is the foundation of many hydrological and biogeochemical processes and plays a decisive role in sustaining ecological functions and services [5]. There is a balanced plant–water relationship in the primary vegetation of desert areas. With the increase in population and social development in desert areas, people's need for forest vegetation ecosystem's goods and services have changed. To meet the increasing needs of people for some special goods and services and diversity that original vegetation cannot provide, more and more original vegetation has been changed into non-native forest land, grassland, and cropland in water-limited areas such as in the Loess Plateau in China [6]. A strategy that plants use to improve survival during seasonal drought is to establish deep roots for their ability to tap into groundwater [1]. Because introduced forest species such as *Robinia pseudoacacia* or *Pinus tabuliformis*, shrub species such as *Caragana microphylla* and grass species such as alfafa have developed extensive root systems that have been reported to reach a depth of more than 1000 cm [7] and more than maximum infiltration depth [3,6,8–10]. Compared with construction species in natural vegetation, the tree species introduced during revegetation consume more water;

therefore, an ecological problem occurs because the regional precipitation cannot meet the evapotranspirative demand of the introduced vegetation [11,12]. Deep-rooted plants such as forest trees not only absorb and use contemporaneous precipitation in shallow soils but also use deep soil water (>100 cm) that has accumulated over the long term [13]. Natural precipitation on the Loess Plateau could not meet the water consumption needs of mature forests, leading to the continuous use of deep soil water [14]. Although the absorption of deep soil water through roots and its utilization by plants can help alleviate water stress in plants [15,16], the excessive consumption of deep soil water on the Loess Plateau has weakened the ecological function of the soil reservoirs and has gradually resulted in soil desiccation, reduced groundwater replenishment rates, and a series of new ecological and hydrological problems [17]. With the growth of plants, plant canopy closure, and root system development, soil drying appeared and sometimes soil drying becomes gradually serious with time in most forest land, grassland, and cropland in desert regions such as in China's Loess Plateau because these desert regions feature low and highly variable seasonal and annual rainfall without irrigation. Serious drying of soil eventually happens in most non-native forest land, grassland, and cropland, and then results in soil degradation, vegetation decline, and agriculture failure. The goal of afforestation is to increase vegetation cover while increasing canopy interception, soil water consumption, yield and services of plant community systems, and changing soil hydrological characteristics such as runoff, soil water consumption, and deep leakage.

Soil water is only used by plant roots, and soil water in most water-limited regions only comes from precipitation. The maximum infiltration depth (MID) and soil water supplies are limited in this region [6,10,13]. Thus, root depth can exceed the depth of soil water recharge from rainwater, leading to severe desiccation of soil in rooting soil layers. Consequently, the combination of increased water use by plants and low water recharge rates has led to soil deterioration, receding vegetation, and crop failure on the Loess Plateau in the perennial artificial grass and forest land [6,18]. Such soil deterioration can adversely affect ecosystem function and services and the stability of manmade forest and vegetation ecosystems, and consequently reduces the ecological, economic, and societal benefits of forest and other plant communities. In turn, this suggests that the relationship between plant growth and soil water (RBPGSW) in these perennial artificial grass and forest lands is not in harmony and should be regulated. In the Loess Plateau of China, the groundwater level remains at a depth of about 20–300 m, precluding any upward capillary flow from groundwater into the root zone [19].

Drought is a recurring natural phenomenon. The complex nature and widespread impact of drought on forest and grass land with high coverage and production—driven by artificial vegetation consuming more than the permissible quantity of soil water resources in water-limited regions—means that regulating the RBPGSW is needed to maintain the SWC of restoring vegetation at levels that sustainably use soil water resources. In order to solve these problems, the plant–water relationship has to be regulated on the -carrying capacity of soil moisture for vegetation (CCSMV) [9,20,21] in key periods of plant–water relationship regulation when the soil water storage in the maximum infiltration depth (MID) is equal to the use limit of soil moisture resources by plants (ULSMRP). The purpose of this paper is to introduce the theory of the ULSWRP and the carrying capacity of soil moisture for vegetation (CCSMV) in order to carry out rational use of soil water resources and promote high-quality and sustainable development of forests, grasslands, and crops in desert regions.

## 2. Materials and Methods

It is very important to study the method of rational utilization of soil water in desert areas because there is little precipitation in these areas and soil water mainly comes from precipitation. In the process of soil water utilization, it is easy to waste soil water resources in wet years or overuse soil water resources in dry years [22–24], which is not good for the

sustainable use of soil water resources and the high-quality sustainable development of forests, grassland, and high-quality production of fruit and crops in desert regions.

### 2.1. Study Preparation

The study area is located in the Shanhuang eco-experiment monitoring station (see Figure 1). The Heici mountain was selected as the study area. Within the non-native forest such as the caragana stand with similar site conditions in the middle of the Heici mountain, five similar 100 m$^2$ (5 m × 20 m = 100 m$^2$) plots with the same conditions, including slope gradient (see Table 1), slope direction, slope position, and vegetation type and soil were prepared. The planting density of the 16-year-old caragana (*Caragana korshinskii*) shrubland was 87 shrubs per 100 m$^2$. One experimental plot kept this density whereas the other four densities were reduced by cutting away the aboveground caragana plants to establish five field experimental plots with different planting densities of 87, 71, 51, 32, and 16 shrubs per 100 m$^2$. The experiment lasted from 2002 to 2020, and the *caragana korshinskii* stems that grew from the base of the cut shrubs were cut when they had grown to a height of about 5 cm to maintain the specified planting density.

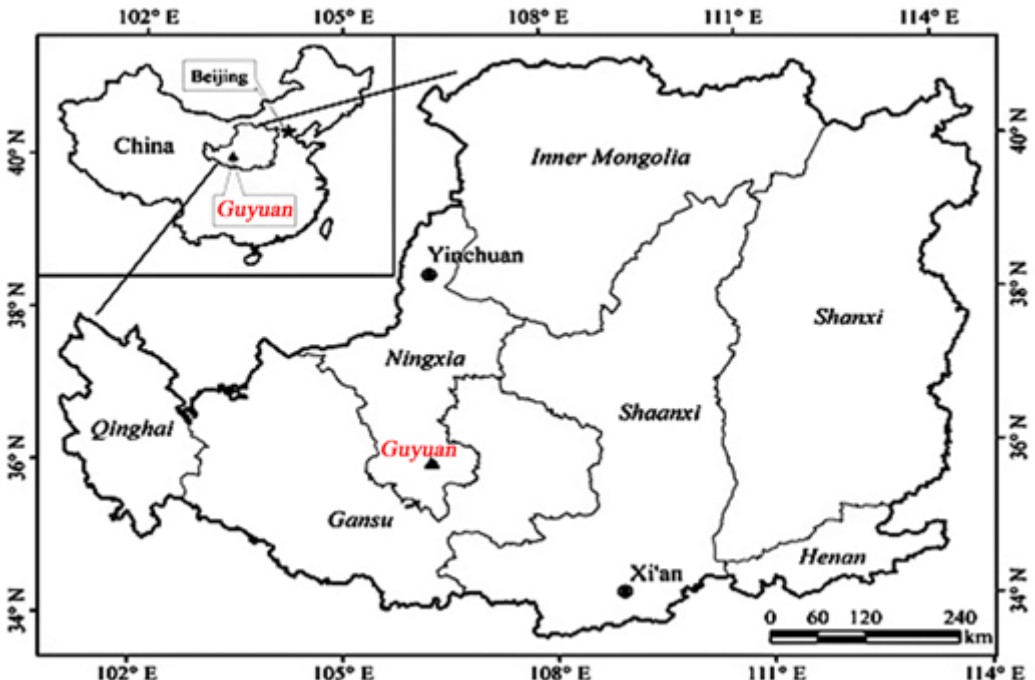

**Figure 1.** The location of the study area on the Chinese Loess Plateau.

**Table 1.** Slope and canopy projection area of *Caragana korshinkii* at different planting densities. (shrubs per 100 m$^2$).

| Planting density | 87 | 71 | 51 | 32 | 16 |
|---|---|---|---|---|---|
| Slope gradient (°) | 7.9 | 7.8 | 8.5 | 7.7 | 7 |
| Projection area (m$^2$) | 99.2 | 99.1 | 99.1 | 99.3 | 99.4 |

Another five similar 100 m$^2$ plots were prepared with the same slope and soil. Different sowing seed amounts were sowed with 0.5, 1.0, 1.5, and 2.0 kg per 100 m$^2$ on 24 June, 2002, to get a young caragana stand. The 1 m × 1 m quadrants were used to investigate the caragana planting density at each plot.

## 2.2. Canopy Interception Measurements

The rainfall data from the study site from 1983 up to 2001 were obtained from the Shanhuang eco-experiment monitoring station, located about 50 m away from the study area. Three standard rain gauges with a diameter of 20 cm were placed in the open space near each plot to measure rainfall outside of the caragana shrubland. An additional 2 to 4 shorter rain gauges with the same diameter were placed under the sample caragana shrubs in the plots to measure throughfall. Canopy interception was measured after rainfall events. The canopy interception (CI) of a single caragana canopy equaled the difference between the rainfall measured in the areas without shrubs ($P_2$) and that measured beneath the canopy of the caragana shrubs ($P_1$); i.e., CI = $P_2 - P_1$. The rate of canopy interception in a plot (Cp) for a specific growth stage is obtained with Cp = CI × CD [6,8,25].

## 2.3. Runoff Measurement

One upper edge and two edges of the plot were enclosed by concrete slabs, 50 cm high and 30 cm deep, forming a 100 m$^2$ (5 m × 20 m) runoff plot. The slope gradient is given in Table 1. Runoff was channeled from the lower end of the plot into a covered collection bin. The height of the water in the bucket was measured and then the runoff was estimated. The runoff was recorded after each rainfall.

$$\mathrm{RF} = \frac{\pi r^2 - p_{2 \times s_i} + 9\pi H R^2}{S_i} \tag{1}$$

where RF is runoff in mm and $P_2$ is rainfall outside shrubland in mm. The symbols $r$ and $R$ represent the radius of the small runoff bucket of 30 cm and the large runoff bucket of 40 cm, respectively; H and $H$ express the height of the small runoff bucket and large runoff bucket, respectively, and $s_i$ and $S_i$ represent the projected area of the collecting flume and the runoff field, respectively.

## 2.4. Root Distribution

A shrub was selected to be used as a sample plant with about mean height and basic diameter of the caragana shrubs near the plant plot. The aboveground parts of the caragana shrub were cut and a hole was dug near the base of the caragana shrub (1 m × 1 m in area and 5 m deep). Average root diameter and biomass were measured for soil depths of 0 to10 cm, 10 to 50 cm, 50 to 100 cm, 100 to 200 cm, 200 to 300 cm, 300 to 400 cm, and 400 to 500 cm.

In each runoff plot, the dry biomass of the herbaceous plants under the shrubs was investigated from samples, which were cut and collected from four 1 m × 1 m quadrant plots and dried at 105 °C, and a visual assessment of the overall coverage area of these plants was conducted. Plots were measured, and the survival rate was estimated the following spring.

## 2.5. Soil Water Measurement

Two 4-meter-long aluminum tubes were inserted into the soil in the center of each test area with a contour distance of 2 m between them. A neutron detector, CNC503A (DR), made by Beijing Nuclear Instrument Company in Beijing, was used to monitor the field soil water content. Neutron detector calibration was performed on the soils of the study site using standard methods [26]. The neutron detector detects slow neutrons, which are linearly related to soil moisture content [27]. The calibration equation is:

$$A = 55.76\,B + 1.89 \tag{2}$$

where A is the percentage of soil volumetric water content (%) and B is the ratio of the neutron count in soil to the standard count in water. The process was begun at a depth of 5 cm and measured at a depth of 4 m every 15 days, increasing by 20 cm. In addition to the 5 cm depth representing 10 cm of the soil surface, the soil water content obtained at

each measured depth represented the soil layer $\pm 10$ cm deep at the measured point. The neutron count lasted 16 s. In addition, we measured soil moisture content before and after rainfall. In addition, soil water content in a red plum apricot plantation was measured from 2017 to 2020, and alfafa grassland from 2011 to 2013.

The two-curve method was used to determine the rainfall infiltration depth of a rainfall event [8,23]. The recharge depth is the distance from the surface of the soil profile to the intersection of the two vertical soil moisture curves before a rain event and after the rain event. There is no groundwater effect, and it is assumed that there is no lateral flow, and the water consumption of planting plants during heavy rainfall is negligible [8,9,25].

The soil moisture supply (SMS) was calculated from:

$$SMS = P_2 - CI - RF \tag{3}$$

where $P_2$ expresses the sum of precipitation outside the shrubs and CI canopy interception and RF expresses runoff (mm).

Based on needs, soil moisture consumption (SMC) was estimated from:

$$SMC = W_1 - W_2 + SMS \tag{4}$$

where $W_1$ expresses the initial soil water resources for a while in mm, and $W_2$ represents the final soil water resources for a while in mm.

The pit was dug in a non-native forest such as caragana, and alfalfa grass samples were collected at the experimental sites for use in investigating soil profiles. The sampling pit dimensions were 1 m$^2$ × 4 m depth, and they were dug in the caragana shrubland in September 2012 and in the alfalfa grassland in 2015. The undisturbed soil samples were collected in triplicate at depths of 0 to 5 cm, 20 to 25 cm, 40 to 45 cm, 80 to 85 cm, 120 to 125 cm, 160 to 165 cm, 200 to 205 cm, 240 to 245, 320 to 325, and 380 to 385 cm with cutting rings (5 cm high, 5 cm inner diameter, and 100 cm$^3$ volume), and the excess soil at the openings on both sides of the ring were cut using a sharp knife, sealed, and transported to the laboratory for use in subsequent analyses.

### 2.6. Moisture Content Measurement at Different Soil Suctions

The cutting ring was used to measure the soil bulk density, total porosity, saturation moisture content, and capillary porosity. The core samples (soil sample undisturbed) collected were used with cutting rings to measure the soil bulk density, noncapillary porosity, and capillary porosity. The bulk density was determined by oven-drying the cores at 105–110 °C, and the total porosity was calculated as 1-bulk density/soil particles density, assuming that the density of soil particles was 2.65 g/cm$^3$. Capillary porosity was equal to the difference between total porosity and non-capillary porosity. Organic content was measured using the potassium dichromate volumetric method. A laser granulometer is commonly used for the analysis of the grain sizes of marine sediments. Soil particle sizes were measured using a master sizer 2000 laser particle analyzer (Malvern Instruments Ltd., Malvern, UK) and grain size was graded based on the United States Department of Agriculture classification system for particle sizes. Soil water concentrations at different soil suctions (0.1, 0.2, 0.4, 0.6, 0.8, 1.0, 2.0, 4.0, 6.0, 8.0 bar, 1 bar = 0.1 MPa) were measured using a Hitachi centrifuge (Hitachi Instruments Inc., Tokyo, Japan). The soil moisture characteristic curve was established to determine the wilting coefficient when soil suction equals 15 bar.

### 2.7. Estimating Method for the Critical Period of Plant–Water Relationship Regulation

When the soil moisture resources in the MID equaled the ULSMRP one day, it is the starting time to regulate the relationship between soil water and plant growth [28,29]. From the starting time, according to the forecasting SWCCV, which was estimated by the precipitation, soil water supply, and single plant consumption in the key period of plant–water relationship regulation, plants A, A = P − VCC, were removed at the maximum

experimental planting density, ensuring the maximum experimental planting density was more than the soil water carrying capacity for vegetation in different situations because the soil water carrying capacity for vegetation changes with precipitation, which is the main source of soil water resources in water-limited regions. The plant–water relationship was continuously adjusted day by day in the same manner, ensuring an even distribution of existing plants. P is the preserved plant density and VCC is the carrying capacity. Then the plant growth and soil water were investigated in the thinned plot and un-thinned plot. If the plant density is adjusted on day I, the plant–water relationship is normal and the plant grows healthily, and if the plant density is adjusted on day I + 1, the plant growth is unhealthy, the vegetation declines, and the crops fail, so the duration of day I is the critical period for the regulation of the plant–water relationship.

## 3. Results and Discussion

### 3.1. Soil Water Resources

Soil water resources are soil water storage in the soil, which are renewable water resources and a component of water resources. The concept of soil water resources emerged in 1986 [30] after the concept of overall soil moistening [31]. There are generalized soil water resources and narrow-sense soil water resources. Generalized soil water resources can be defined as the water stored in the soil from the surface soil to the water table, commonly used in geology or architecture, and narrow soil water resources are the soil water storage in the root zone soil, commonly used in forestry, grass, and agriculture. Soil water resources are renewable water resources and are components of water resources [25,32].

In addition, there are dynamic soil water resources, which are the antecedent soil water resources plus the soil water supply from precipitation in the critical period of plant–water relationship regulation, the growing season for deciduous plants, or over a year for evergreen plants [32]. Soil water resources change with rainfall, runoff, soil evaporation, plant transpiration, deep leakage, and soil water movement.

### 3.2. Root Water Absorption

Roots are the most important organ for terrestrial plants to suck soil water even though stoma in leaves, and a stem can absorb a little water when air humidity is high, such as during rain. So, root distribution is the most important index to characterize soil water deficit.

Root distribution can be investigated by the soil-pit method. For example, most of the caragana root biomass is distributed in the 0 to 200 cm soil layer even though the root extends to 5.0 m in 16-year-old *caragana korshinskii* shrubs in the semi-arid hilly region of the Loess Plateau (Guyuan, China), see Figure 2 [6], which is more than the maximum infiltration depth.

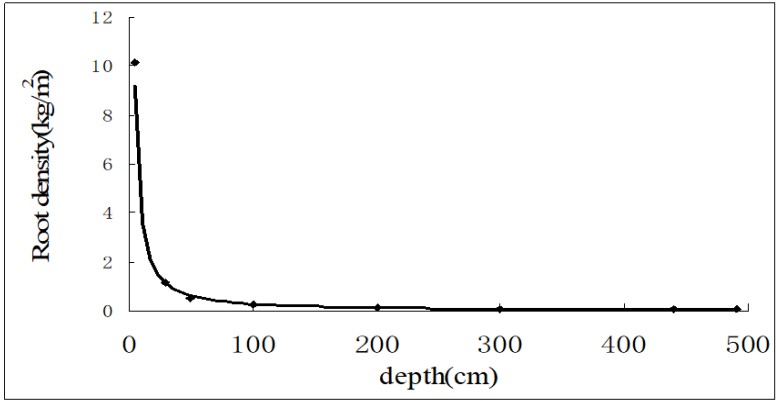

**Figure 2.** Root distribution of 16-year-old *Caragana korshinskii* shrubs in the semi-arid hilly region of the Loess Plateau.

The soil water can be used by plants, but plants do not continue to absorb soil water. If the soil moisture supply is less than the soil moisture consumption, the soil water resources will reduce to lethal soil water resources (LSWR), which is the soil water storage when soil water content in the root zone soil is equal to the wilting conference. When soil moisture resources equal the LSWR and there is no timely soil moisture supply from rain, the plant will die, because at that time the available soil moisture for the plant in the root zone soil is used up.

### 3.3. Use Limit of Soil Moisture Resource by Plants (ULSMRP)

Soil water can only be used by plant roots, and the use of soil water resources by plants must be limited because soil water mainly comes from precipitation and is limited with big seasonal and yearly changes, and soil moisture only come from precipitation in most water-limited regions. The control degree can be expressed by the use limit of soil moisture resources by plants (ULSMRP) [25,33]. SWRULP is the use limit of soil moisture resources by plants, which can be defined as the soil moisture storage in the maximum infiltration depth (MID), at which the soil moisture content is equal to the wilting coefficient. The wilting coefficient is expressed by the wilting coefficient of the indicator plant because the wilting coefficient changes with different plants, and many single species of plants live together to form a plant population or many different species of plants live together to form a plant community and then use solar radiation and soil moisture to produce goods and services. Indicator plants of natural vegetation are established species, and indicator plants of non-native trees or grasses for man-made forest and grass land are principal or purpose species. For example, the indicator plant of caragana shrubland is caragana even if there are some other species of herbaceous plants growing under the shrubs, including *Heteropappus attaicus*, *Stipa bungeana*, *Lespedeza davurica*, *Artemisia giraldii*, and *Thymus mongolicus* [6,25].

Soil moisture resources can be defined as the soil moisture storage in the root soil zone, which is the best indicator to express soil moisture state because plant roots are distributed vertically in the root soil zone and absorb soil moisture from the root soil zone. Along with plants absorbing soil water, soil water content reduces in the root soil zone, moisture stress increases, and the soil moisture resources and the power of plant self-regulation through stomatal opening are limited. Therefore, the degree to which plant roots absorb soil moisture must be limited in most moisture-limited regions. The limit is expressed by the use limit of soil moisture resources by plants (ULSMRP) [33].

ULSMRP can be defined as the soil moisture storage in the MID in which the soil moisture content of each soil layer is equal to the withering coefficient. The wilting coefficient of a plant community is expressed by the wilting coefficient of the indicator plant. Some plant communities are formed by a single plant species and others are formed by many plant species living together and forming a community to use space resources, soil water resources, or soil nutrient resources. Although any one species in a plant community can express ULSMRP or CCSMV in this situation in theory, the different plant species differ in their positions and roles in the community. Constructive species for natural vegetation and principal or purpose species of trees or grasses are selected as drought-resistant plant species. Principal species are the main afforestation tree species and account for the majority of the trees planted in a region, and purpose species of trees or grasses are those cultured or managed by people in a region, such as *salix mongolica* and *Haloxylon ammodendron*. Generally, principal species of trees or grasses are also purpose species in a region, and corn is a purpose species in cornland because it helps with weeding and plastic mulching.

### 3.4. Infiltration and Maximum Infiltration Depth

Maximum infiltration depth is the most important indicator for estimating the soil moisture resources use limit by plants. Infiltration depth includes the infiltration depth for one rainfall and maximum infiltration depth because there is cumulative infiltration, such as in China's Loess Plateau, because of loess clay adsorption and continuous rainfall. If you investigate the change in soil moisture content with soil depth before a rain event

according to the weather report and the change in soil moisture content with soil depth after the rain event, the infiltration depth for one rain event can be determined by the two-curve method—the infiltration depth is equal to the distance from the land surface to the crossover point between the two adjacent vertical soil moisture distribution curves with soil depth before a rain event and after the rain event, see Figure 3. The MID will occur after a continuous heavy rainfall event or in a long-term infiltration process and can be determined by a series of two-curve methods, see Figure 4 [8,23,25]. Under natural conditions, there is a maximal infiltration depth. Even generally, root distribution depth in water-limited regions is more than the maximal infiltration depth and takes moisture from the soil layers below the wetting front depth, but soil moisture mainly from precipitation and water cannot be recovered once the soil moisture content in the soil layers below the wetting front depth is smaller than the wilting coefficient, which will influence plant growth severely and cause vegetation decline. The SWRULP was 222.8 mm and the LSWR was 405.7 mm in the 16-year-old caragana shrubland in the semiarid hilly Loess region of China.

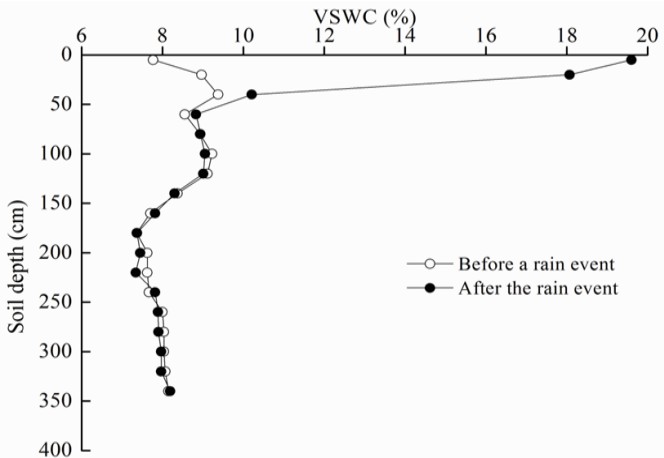

**Figure 3.** The vertical changes in soil moisture with increasing soil depth after and before a rainfall event in *caragana korshinskii* shrubland.

Infiltration depth after and before a rainfall event can be determined by the two-curve method discovered in 2003 and named in 2015. The infiltration depth is equal to the distance from the surface to the crossover point between the two adjacent soil moisture distribution curves of soil moisture with soil depth before and after the rain event (see Figure 3).

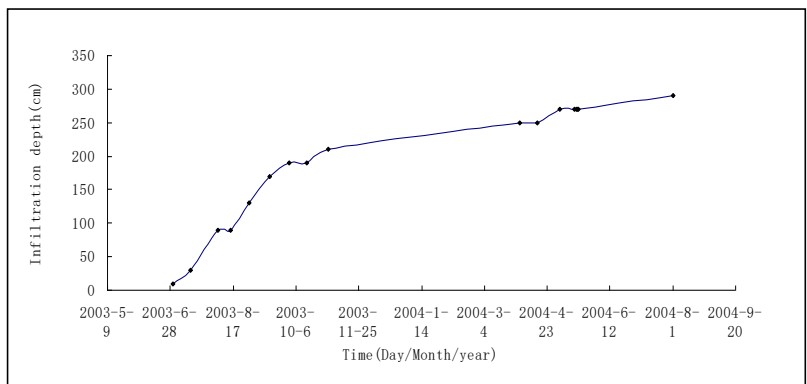

**Figure 4.** Change in the cumulative infiltration process with increasing time in *caragana korshinskii* shrubland in the semiarid and hilly Loess region (Guyuan, China).

In nature, the infiltration process, including the MID, occurs after a continuous heavy rainfall event and a long-term cumulative infiltration process, and can be determined by a series of two-curve methods (see Figure 4). The maximum infiltration depth was 290 cm (see Figure 4) in the 16-year-old caragana and red plum apricot plantation [10,23,25].

### 3.5. Carrying Capacity

Natural resources are limited, which limits the maximum size that a population can safely obtain in desert soil. Carrying capacity is the best indicator to express the creature–environment relationship and is the core issue for high-quality and sustainable development.

After "sustainable development" (1987), the term "carrying capacity" was first used by range managers [34] and U.S. Department of Agriculture researchers [35], following the 1920 formulation of the logistics equation by Raymond Pearl and Lowell J. Eugene Odum (1953), which relates the carrying capacity term to the constant k (see Equation (1)):

$$\frac{dN(t)}{dt} = rN(t)(\frac{K - N(t)}{K})\ t\ >\ 0$$

where $N(t)$ is the density at time $t$, the population per unit area at time $t$; $r$ is the intrinsic growth rate; $r > 0$; and $K$ is an asymptote (the carrying capacity), where $K > 0$ (Guo 2019).

The maximum plant population can be expressed by carrying capacity, which is the maximum plant population that nature resources can carry.

### 3.6. Carrying Capacity of Soil Water for Vegetation

Vegetation carrying capacity is the ability of land to support vegetation. Vegetation includes different plant communities that changes with plant species. To solve serious drying of soil—because the serious drying of soil eventually results in soil quality degradation, vegetation quality decline, and crop failure—the concept of carrying capacity of soil moisture for vegetation appeared.

The ability of soil moisture resources to carry vegetation is limited because soil water resources are limited. The limitation can be expressed by the carrying capacity of soil moisture for vegetation (CCSMV). The CCSMV first appeared in 2000 [20]. It can be defined as the highest density or population quantity (absolute index) of indicator plants in a plant community when soil moisture consumption equals soil moisture supply in the root zone soil in a given period, especially the key period of regulating the plant–water relationship, which is the minimum death day. SWCCV can be determined by the classical carrying capacity model, that is to say, CCSMV is equal to available soil water resources divided by the single plant moisture demand or the soil water–plant density model. We established a series of density experiments in the same conditions and the same plot areas and measured the precipitation, throughfall, runoff, and deep leakage and estimated the soil moisture supply and soil moisture consumption under different density in key periods of regulating between soil moisture and plant growth, and then established the equations between plant density and soil moisture supply and soil moisture consumption, respectively. Soil moisture supply reduces with increased planting density, and the relationship between soil moisture consumption and population quantity or density can be described with a parabolic equation (Figure 5). CCSMV can be estimated by combining and solving the equations simultaneously [9,25].

### 3.7. The Key Period of Regulating the Plant–Water Relationship

The plant–water relationship in a growing period or year can be divided into different periods. The first stage ranges from sowing or dormancy to germination, which is the period of insensitivity or dormancy. After preparing the ground, fertilizing the fields, and planting or sowing seeds, the soil water content is high and the plant–water relationship is good. At this point, soil moisture is not the limiting factor for plant growth. The second stage is the period from germination to full expansion of the leaves. At this point, the force of soil moisture on plant growth increases. When the soil moisture resources in the MID are

equal to the SWRULP, it is called the starting time to consider regulating the relationship between plants and water [28]. Then the plant–water relationship goes into a key period of regulating soil moisture and the plant-growth relationship mast be paid more attention to. The key period of plant–water relationship regulation is the most important period, which decides the maximum output and benefit of the forest and vegetation. If the planting density is more than the CCSMV in this period, the relationship between soil water and plant growth must be regulated based on the CCSMV, especially the CCSMV in the key period of regulating the relationship between soil water and plant growth.

The key period of regulating the relationship between soil water and plant growth can be determined by the thinning method. We can establish the maximal planting density, that is, the planting density of the indicator plant when the precipitation is at the maximum because the precipitation in water-limited regions changes greatly. For example, the precipitation of the Guyuan eco-experiment station ranged from 284.3 mm in 1986 to 634.7 mm in 1984 in the semiarid hilly Loess region [6,25].

When the soil moisture resources in the MID equal the ULSWRP on a given day, it is time to consider regulate the relationship between soil water and plant growth [25,28]. From that time, according to the forecasting CCSMV estimated by the precipitation, soil water supply, and single plant consumption in the key period of plant moisture relationship regulation, plants A, A = P − VCC, were removed at the maximum experimental planting density and the plant- moisture relationship was continuously adjusted day-by-day in the same manner. P is the preserved plant density and VCC is the CCSMV. Then the plant growth and soil water were investigated in the thinned plot and un-thinned plot. If the plant density is adjusted on day I, the plant–water relationship is normal and the plant grows healthily, and if the plant density is adjusted on day I + 1, the plant growth is unhealthy, which causes the vegetation to decline and the crops to fail. The duration of day I is the critical period for the regulation of the plant–water relationship. If the soil water resources were more than the SWRULP because of a rain event, the experiment was stopped, which shows that the plant–water relationship did not need to be regulated in the year.

If the rainfall occurs at the critical period of the plant–water relationship, the surface runoff, before and after the rain, the soil water consumption, and the deep leakage are observed; soil water content at different soil depth; and then estimate soil moisture consumption and soil water supply amount; the forecasted SWCCV can be tested according to the relationships between plant density and soil moisture supply; and the relationship between plant density and soil moisture consumption in the key period of regulating between soil moisture and plant growth that can be assessed. The equation to solve include the carrying capacity of soil moisture for vegetation in the positive root. The changes in soil water supply, soil moisture consumption, and seed yield with plant density see Table 2, Figure 5 and Figure 6 [6,8,9,23–25].

**Table 2.** The changes in soil water supply, soil moisture consumption, and seed yield with plant density.

| Plant density (Brushes/100 m$^2$) | 87 | 71 | 51 | 32 | 16 |
|---|---|---|---|---|---|
| Soil water supply (mm) | 69.64 | 68.53 | 78.11 | 82.35 | 88.97 |
| Soil water consumption (mm) | 88.57 | 68.28 | 63.58 | 45.42 | 58.4 |
| Seed yield (g/100 m$^2$) | 39 | 76 | 130 | 63 | 45.5 |

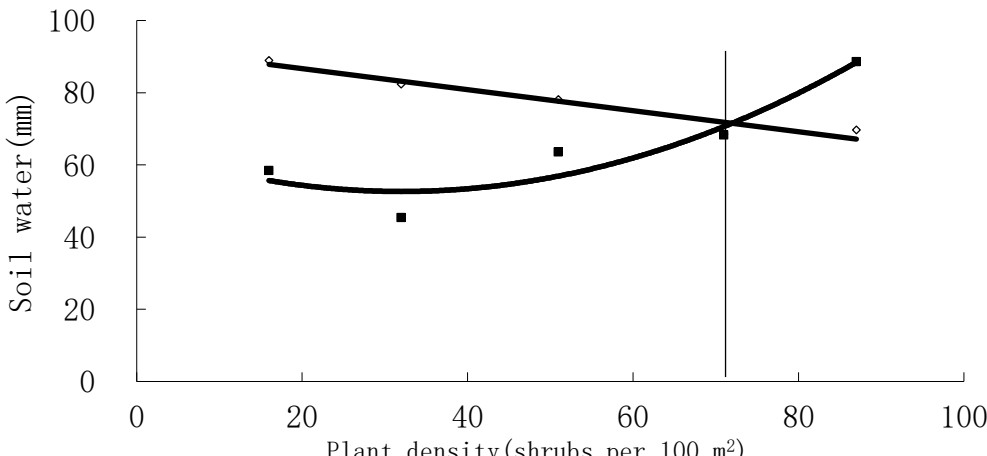

**Figure 5.** The changes in soil moisture supply (hollow circle) and soil moisture consumption (solid square) with plant density in the key period of the plant moisture relationship regulation of 18-year-old caragana shrubland in the semiarid hilly Loess region. The cross-point is the carrying capacity of soil water for vegetation.

The maximum seed yield can be obtained by further regulating plant density (see Figure 6) because the plant density (PD) and seed yield (SD) relationship can be expressed by $SD = 106.6e^{(-0.0008pd^2)}$ [29].When the plant density is about 51 shrubs per 100 m², the seed yield is close to the maximum yield. The results indicate that the maximum planting density of *Caragana korshinskii* as economic forest was less than the soil water carrying capacity for vegetation. In order to get the maximum yield and services, if the duration of severe soil drying is more than the key period of plant water relation regulation, we should further regulated the relationship between reproductive growth and nutrient growth because the plant water relation of some plantation is regulates by pruning some branches or leaves according to suitable branches and leaves when plant density is equal to CCSMV, instead of cutting some tree, such as fruit tree or crop, and another plant by cutting some plant on the relationship between economic yield and plant density.

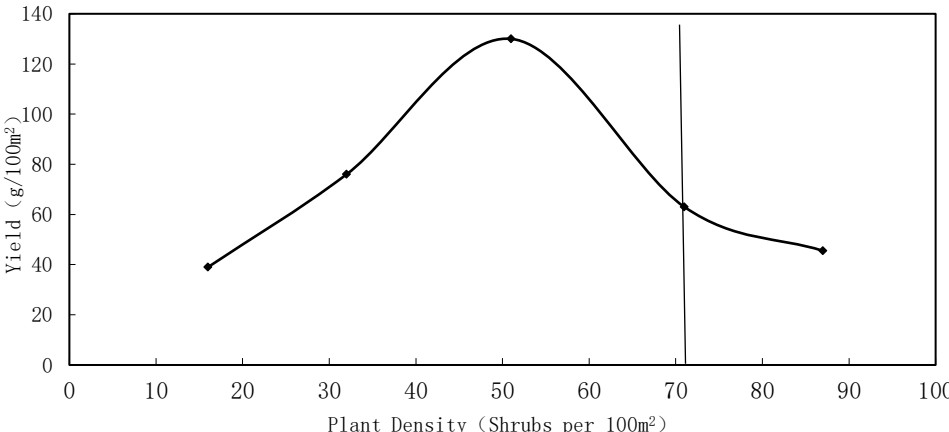

**Figure 6.** The relationship between plant density, carrying capacity, and caragana seed yield in the semiarid hilly Loess region.

## 4. Discussion

Soil water resources are water reserved in the soil. In most desert regions, soil water resources are the most important and precious resources that influence plant growth and obtain maximum yield and service. There is a balanced plant–water relationship in the primary vegetation of desert areas. With the increase in population and social development in desert areas, people's need for forest vegetation ecosystems' goods and services have

changed and strengthened. To meet the increasing needs of people for special goods and services and the diversity that original vegetation cannot provide, more and more original vegetation has been changed into non-native forestland, grassland, and cropland in water-limited areas such as in China's Loess Plateau [6], and many foreign species are introduced into desert regions. Foreign species develop into different vegetation types and then change some hydrological characteristics, such as canopy interception, runoff, soil infiltration, and deep leakage, which change soil water supply and soil water consumption. As s strategy, most plants develop and establish deep roots for their ability to tap into groundwater [1]. Because underwater is deep and without irrigation, soil water resources mainly come from precipitation [6,9,25]. Because precipitation is small and limited, soil water resources are limited in most of desert regions. In order to realize high-quality sustainable development of forest vegetation and high-quality production of fruit and crops, we should closely care about the water–plant relationship because climate changes over the years and there is a maximum yield and service every year. If abnormal phenomena of the water–plant relationship, such as plant overuse of soil water resources or wasted soil water resources, happens in the key regulation period of the plant–water relationship, we should regulate the water–plant relationship in terms of the carrying capacity of soil water for vegetation in the key regulation period of the plant–water relationship and obtain the maximum yield and beneficial results and realize a sustainable use of soil moisture resources.

## 5. Conclusions

There is a balanced plant–water relationship in the primary vegetation of desert areas because the plant conductivity, canopy interception, and soil moisture use depth and soil moisture consumption are low, and the soil moisture supply is bigger. There is no serious soil drought or soil degradation. With the increase in population and social development in desert areas, people's need for forest vegetation ecosystems' goods and services have changed. To meet the increasing needs of people for special goods and services and the diversity that original vegetation cannot provide, more and more original vegetation has been changed into non-native forestland, grassland, and cropland in moisture-limited regions, which increase canopy interception, soil water use depth, and soil water consumption and change soil hydrological characteristics, but non-native forestland, grassland, and cropland have a lower capacity to regulate the soil water and plant growth relationship, which results in soil degradation, vegetation decline, and crop failure in dry years or the waste of soil water resources in wet years. Therefore, the plant–water relationship must be regulated in terms of the carrying capacity of soil water for vegetation when the soil moisture resources in the maximum infiltration depth equal the use limit of soil moisture resources by plants and the duration of severe soil drying is more than the key period of plant–water relationship regulation. This action not only solves soil quality degradation, vegetation decline, and crop failure but also ensures the sustainable use of soil moisture resources, helps realize high-quality and sustainable management of forest vegetation, and helps produce sustainable fruit trees and crops. For example, In the semiarid hilly Loess region of a 17-year-old caragana forest, serious soil drought and soil quality degradation occurred, leading to the early defoliation of caragana because the plant density of 87 plants per 100 m$^2$ was greater than the soil water-bearing capacity of 72 bushes per 100 m$^2$. When the rainy season came, the canopy interception disappeared, which shows that the plant density that exceeded the soil moisture carrying capacity for vegetation and affected the biggest benefit of *caragana Korshinskii* forest for soil and water conservation before the rainy season appeared on 5 August 2003.

Soil water resources are water stored in soil space, and it is a part of water resources. Only plants can take advantage of soil moisture, otherwise soil water resources will be wasted. Non-native vegetation changes the soil hydrology process and plant–water relationship, and sometimes soil drying happens and becomes gradually serious in non-native vegetation. Serious drying of soil eventually results in soil-quality degradation, vegetation decline, and crop failure, which influence the production and supply of forest

vegetation products and services on the market. Soil moisture resources are valuable resources and must be used in a sustainable way in desert regions. If soil moisture resources ae lower than the use limit of soil moisture resources by plants, the water–plant relationship enters the key period of plant–water relationship regulation. If the duration of severe soil drying is more than the key period of plant–water relationship regulation, the plant–water relationship must be regulated in terms of the carrying capacity of soil water for vegetation in the key regulation period of the plant–water relationship in order to carry out the sustainable use of soil moisture resources, high-quality and sustainable management of forest and grass, and high-quality production of fruit and crops in desert regions.

**Funding:** This research was funded by the National Science Fund of China (Project Nos. 42077079, 41071193, and 41271539) and a study on high-quality sustainable development of soil and moisture conservation (A2180021002).

**Institutional Review Board Statement:** I exclude this statement if the study did not involve humans or animals.

**Informed Consent Statement:** Informed consent was obtained from all subjects involved in the study.

**Data Availability Statement:** Data available on request due to privacy/ethical restrictions.

**Conflicts of Interest:** There is not Competing Financial and non-financial interests.

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
