# Peer review of "Soil Hydrology Process and Rational Use of Soil Water in Desert Regions"

_water, doi:10.3390/w13172377_

Round 1

Reviewer 1 Report

In this revised manuscript, author improved the all the sections at some extent, especially "Materials and Methods" and "Results and Discussion" sections. However, still this manuscript need to modify to make maintain the standard for the publication. Some key points are: 

References need to be formatted according to MDPI guidelines.

Acronyms are need to edit carefully. For example, at line 85 and 89, are these same? if so need to adjust to clarify the message. On the same paragraph, ULSWRP should be ULSMPRP (line 89).  Another example is, MID already explained in line 67 and mentioned again in Line 87

Line 25-26-> provide references

Lines 80-91-> the last paragraph of the Introduction should contain the closing statement why this study was conducted and what were the objectives and what methedology was used to obtain that. 

Line 93-98-> why is that statement on the “Materials and Methods” section? Line 94-97->need to rewrite

Author Response

Dear Editor

We would like to thank you for the opportunity to revise our mannuscript, titled “Soil hydrology process and Sustainable Use of Soil Water Resources in Desert Regions”. We would also like to thank the reviewers for their work and for their helpful suggestions and comments.

Your sincerely,

Dr. Guo Zhong-Sheng

Extended English improvements and formatting on MDPI template

Answer:improved

Abstract improvement, with presented results

Answer:improved

Develop M&M more, add a figure with the study area

Answer:added a figure with the study area

Also, add a methodological workflow

Answer:added a methodological workflow

Better describe the importance of your research and its need in the study area

Discussion chapter missing, further develop and add citations and discussions between your conclusions and other researchers

Answer:improved

Reviewer 2 Report

water-1264437

The manuscript “Soil hydrology process and Sustainable Use of Soil Water Resources in Desert Regions” addresses an interesting and up-to-date subject, which adheres to Water journal policies, but it needs further work before considering for publication.

In my opinion, the manuscript needs the following improvements:

  • Extended English improvements and formatting on MDPI template
  • Abstract improvement, with presented results
  • Develop M&M more, add a figure with the study area
  • Also, add a methodological workflow
  • Better describe the importance of your research and its need in the study area
  • Discussion chapter missing, further develop and add citations and discussions between your conclusions and other researchers

Author Response

(The authors gave the same response as above.)

Round 2

Reviewer 2 Report

The author did small improvements, and the manuscript is still not on the correct template and everything is unorganized. I don’t know if this is deliberate and maybe he expects MDPI publishing editors to do this, even though 2 major revisions have passed.

I will let the Academic Editor to decide.

This manuscript is a resubmission of an earlier submission. The following is a list of the peer review reports and author responses from that submission.

Round 1

Reviewer 1 Report

water-1264437

The manuscript “Soil hydrology process and Sustainable Use of Soil Water Resources in Desert Regions” addresses an interesting and up-to-date subject, which adhere to Water journal policies, but it need allot of further work before it has publishing potential.

The overall manuscript feels rushed, with weak chapters and full of citations problems, and not on the MDPI template and guidelines.

The overall results and discussions feel lacking and weak, with predictable conclusions and no beneficiary to the research.

Also, the M&M chapter was underdeveloped.

While I appreciate the work of a single author, In my opinion this needs further work before resubmission.

Some advice for further improvements:

  • Develop M&M more, add a figure with the study area
  • Also, add a methodological workflow
  • Better describe the importance of your research and it’s need in the study area
  • Discussion chapter was weak, further develop and add citations and discussions between your conclusions and other researchers
  • Add more citations! Such study needs at least 40 citations

Reviewer 2 Report

The introduction is lacking sufficient statistics and references which is essential for strong background and justifies the problem statement and study objectives. Need to discuss more about the current ongoing water-vegetation problem in the research area.

The methodology also needs to improve. Equations used for SWS should describe at the beginning and which measures were taken need to describe consequently. How many sample data were collected?

The results section should have outlined as “Results and Discussions”. I am seeing more discussion rather than “results presentation” in this section.

Subsection 3.3- I am not sure how it is fit in the Results section. This should move to the Introduction or in the Discussion section where the author can explain along with the results. Same comment for subsection-3.5 and 3.6. Please place them in the right place.

Need to use correct citation format according to MDPI rule. I am also seeing lots of self-citations (if not, citations from the same author)

Line 12- 13-> Soil drying or soil moisture depletion? What is serious drying? Please use scientific terms for that

Lines 8-11 and 25-30 are exactly the same! Please rephrase

Line 37-39- give some statistics and references

Provide a table for the data collection

Figure 4- Only 5 samples were collected? Or these are average for each plant density?

Figure 5-> What is the Key message from figure 5? Wasn’t described at all.

Line 384-385-> Could you explain more? Are you making this conclusion from Figures 4 and 5?

Line 385-386-> Not clear what is meant.